# A New Derivative of Retro-2 Displays Antiviral Activity against Respiratory Syncytial Virus

**DOI:** 10.3390/ijms25010415

**Published:** 2023-12-28

**Authors:** Adrien Le Rouzic, Jenna Fix, Robin Vinck, Sandrine Kappler-Gratias, Romain Volmer, Franck Gallardo, Jean-François Eléouët, Mathilde Keck, Jean-Christophe Cintrat, Julien Barbier, Daniel Gillet, Marie Galloux

**Affiliations:** 1INRAE Unité de Virologie et Immunologie Moléculaires (VIM), Université Paris-Saclay-Versailles St Quentin, 78350 Jouy-en-Josas, France; adrien.le-rouzic@inrae.fr (A.L.R.); jenna.fix@inrae.fr (J.F.); jean-francois.eleouet@inrae.fr (J.-F.E.); 2CEA, Département Médicaments et Technologies pour la Santé (DMTS), SIMoS, Université Paris-Saclay, 91191 Gif-sur-Yvette, France; robin.vinck13@gmail.com (R.V.); mathilde.keck@cea.fr (M.K.); julien.barbier@cea.fr (J.B.); 3CEA, Département Médicaments et Technologies pour la Santé (DMTS), SCBM, Université Paris-Saclay, 91191 Gif-sur-Yvette, France; jean-christophe.cintrat@cea.fr; 4NeoVirTech SAS, 1 Place Pierre Potier, 31000 Toulouse, France; skappler@neovirtech.com (S.K.-G.); fgallardo@neovirtech.com (F.G.); 5INRAE, IHAP, UMR 1225, ENVT, 31300 Toulouse, France; romain.volmer@envt.fr

**Keywords:** respiratory syncytial virus, Retro-2.2, fusion protein, retrograde transport, antiviral

## Abstract

Human respiratory syncytial virus (hRSV) is the most common cause of bronchiolitis and pneumonia in newborns, with all children being infected before the age of two. Reinfections are very common throughout life and can cause severe respiratory infections in the elderly and immunocompromised adults. Although vaccines and preventive antibodies have recently been licensed for use in specific subpopulations of patients, there is still no therapeutic treatment commonly available for these infections. Here, we investigated the potential antiviral activity of Retro-2.2, a derivative of the cellular retrograde transport inhibitor Retro-2, against hRSV. We show that Retro-2.2 inhibits hRSV replication in cell culture and impairs the ability of hRSV to form syncytia. Our results suggest that Retro-2.2 treatment affects virus spread by disrupting the trafficking of the viral de novo synthetized F and G glycoproteins to the plasma membrane, leading to a defect in virion morphogenesis. Taken together, our data show that targeting intracellular transport may be an effective strategy against hRSV infection.

## 1. Introduction

The human respiratory syncytial virus (hRSV) is the leading cause of severe lower respiratory tract infection in young children worldwide and the first cause of hospitalization of newborns under 6 months of age [1,2,3]. A systemic multisite study has shown that hRSV is the first etiological agent responsible for severe pneumonia (more than 30%) in hospitalized children in Asia and Africa [4]. It is noteworthy that hRSV is also a frequent cause of otitis in infants and that children who suffer from severe hRSV infection are at risk of developing further respiratory complications such as asthma [5,6]. Reinfections occur throughout life with symptoms of common cold in healthy adults but lead to severe lower respiratory infections in immunocompromised and elderly people [7]. Despite the high impact of hRSV infections on public health, the first vaccines to prevent severe hRSV disease in the elderly were approved by the Food and Drug Administration only in 2023 (GSK, Pfizer, Hong Kong, China) [8,9], followed by a vaccine dedicated to pregnant women (Pfizer) [10]. The only specific treatment commercialized for newborns remains humanized monoclonal antibodies directed against the fusion protein F responsible for viral entry. The first commercialized antibody (Palivizumab, Synagis^®^) displays a poor benefit–cost ratio, limiting its use to prophylaxis of severe infection in preterm and high-risk infants [11,12]. The new prophylactic antibody targeting the pre-fusion form of F (Nirsevimab, Beyfortus^®^) has been very recently approved in the European Union for the prevention of hRSV lower respiratory tract disease in newborns and infants during their first hRSV season [13,14].

hRSV belongs to the order *Mononegavirale*s and the family *Pneumoviridae,* which are enveloped viruses with a single-strand negative sense RNA genome [15,16]. The attachment of the viral particle to the cell surface and the entry into the host cell are mediated by the viral glycoproteins G and F. More specifically, virus entry involves the fusion between viral and cellular membranes. This fusion is pH-independent [17], mediated by F, and could occur at the plasma membrane or in endosomes after macropinocytosis [18]. Of note, in cell culture the F protein is responsible for the fusion between cells, leading to the formation of syncytia which characterizes hRSV infection [19,20]. The fusion allows the release into the cell cytoplasm of the ribonucleoprotein complex (RNP), composed of the viral genome encapsidated by the nucleoprotein N, the viral polymerase L, its cofactor the phosphoprotein P, and the M2-1 protein (transcription factor) [21]. Transcription and replication of the viral genome are initiated by the viral polymerase L, which is responsible for all RNAs synthesis activities (RNA-dependent RNA polymerase, mRNAs capping and methyltransferase). These steps take place in viral factories that are membraneless liquid organelles, called inclusion bodies (IBs) [22,23]. The viral mRNAs are then exported from IBs to the cytoplasm where viral proteins are expressed. At the final stages of the viral cycle, RNPs and structural viral proteins are addressed to the plasma membrane where assembly occurs, leading to the formation of new filamentous virions which are released by membrane fission.

To date, most developed antiviral strategies aim at targeting the viral F protein to impair virus entry or the viral polymerase L [24,25,26,27]. However, none of these compounds have been commercialized yet, with the main limitation being the emergence of escape mutants upon treatment. In this context, the alternative of targeting cellular pathways involved in the viral cycle may represent a new antiviral approach.

Here, we assessed the potential antiviral activity against hRSV of Retro-2.2, a new optimized derivative of the cellular retrograde transport inhibitor Retro-2 [28]. Retro-2 was initially identified as an inhibitor of the ricin toxin [28]. This molecule and its optimized derivative Retro-2.1 were subsequently shown to display an antiviral activity against various viruses such as polyomaviruses and papillomaviruses [29,30,31], adeno-associated viruses [32], poxviruses [33], Ebola and Marburg viruses [34], and more recently SARS-CoV-2 [35]. Here, our results reveal that hRSV replication is inhibited by Retro-2.2 treatment in infected cell cultures. The treatment induced a defect in the formation of syncytia among hRSV-infected cells that correlated with a defect of virions release. Further investigation revealed that Retro-2.2 affects the trafficking of the viral membrane proteins F and G to the plasma membrane. Altogether, our results highlight a new antiviral approach against hRSV, based on the inhibition of the final stages of virion assembly and release.

## 2. Results

### 2.1. Retro-2.2 Antiviral Activity against hRSV

Following in vitro screening of Retro-2.1 and different optimized Retro-2 analogues (EP3886859) on DNA and RNA viruses, we investigated the antiviral activity of the Retro-2.2 molecule (Figure 1A) against hRSV virus.

The antiviral activity of Retro-2.2 against hRSV was first assessed on HEp-2 cells infected with a recombinant hRSV expressing the mCherry reporter gene (mCherry-rRSV), which allows to quantify the inhibitory activity of compounds by fluorescence measurement [36]. Cells were infected with mCherry-rRSV either in the absence or the presence of serial dilutions of Retro-2.2 for 2 h. After infection, the medium was replaced by medium containing serial dilutions of Retro-2.2, and fluorescence intensity was measured 48 h post-infection. In parallel, the potential cytotoxicity of Retro-2.2 in the same conditions was evaluated on non-infected cells. As shown in Figure 1B, similar results were obtained when the cells were treated during or only after infection, with half-maximal inhibitory concentration (IC_50_) close to 1.6 µM. Furthermore, pre-treatment of cells for 3 h in the presence of Retro-2.2 before the infection did not increase the efficiency of Retro-2.2 antiviral activity (Appendix A). These results suggest that Retro-2.2 has no impact on viral particles or on the early steps of virus entry. However, in our conditions, a cytotoxic activity of Retro-2.2 was observed, with a calculated half-maximal cytotoxic concentration (CC_50_) close to 15 µM. We further investigated the cytotoxicity of Retro-2.2 on cultures with different cell confluence. Our results suggested that non-confluent cells are less sensitive to Retro-2.2 treatment than confluent cells (Appendix A). The impact of Retro-2.2 on hRSV replication was also validated on different cell lines usually used as a model for hRSV studies, i.e., A549, BEAS-2B, and 293T, at a concentration of 3 µM (Appendix A). We then further evaluated the effect of Retro-2.2 treatment on viral shedding by quantifying virions in the culture medium of infected cells, by plaque formation assay. Our results showed that treatment with 2 µM Retro-2.2 induces a log_10_ decrease in viral titer compared to untreated cells, as well as a reduction of plaque size (Figure 1C). A direct observation of infected cells by fluorescence also revealed a strong effect of Retro-2.2 treatment on the size of the syncytia formed during hRSV infection, with the presence of large syncytia in the control condition compared to numerous but small syncytia in the presence of Retro-2.2 (Figure 1D).

Altogether, these results show that Retro-2.2 inhibits hRSV replication in cell culture. This inhibition is characterized by a defect of syncytia formation and of viral particles release.

### 2.2. Retro-2.2 Effect on the Formation of IBs and Virions Assembly in hRSV-Infected Cells

In order to characterize the inhibitory effect of Retro-2.2 against hRSV, we then studied the impact of Retro-2.2 treatment on the formation of IBs and on viral assembly. BEAS-2B cells, which are usually used for the study of hRSV replication by immunofluorescence, were infected with recombinant hRSV expressing the firefly luciferase (Luc-rRSV), which displays the same hRSV sequence and kinetics of replication compared to mCherry-rRSV [36], at MOI 0.2. Cells were incubated for 36 h in the absence or presence of 2 µM Retro-2.2, a concentration close to the IC_50_. The N and F proteins were then immunolabeled in order to observe IBs and virions assembly, respectively. Within syncytia observed in untreated cell cultures, the F protein was concentrated in granules around the nuclei, but also at the plasma membrane where a large number of short filaments were observed (Figure 2A). Similar localization of F was observed in cells treated with Retro-2.2, although the distribution of F appeared less homogenous along the plasma membrane, and longer filaments were detected compared to the control condition. However, N labeling allowed to validate the presence of IBs of similar shape in both conditions, suggesting that Retro-2.2 did not impair viral genome transcription or replication, nor viral proteins expression (Figure 2A).

Although the formation of syncytia in untreated cell culture makes the comparison difficult, these results suggest that Retro-2.2 could affect the activity of the F protein. We therefore studied the effect of Retro-2.2 in the context of cells expressing only the F protein. BEAS-2B cells were transfected with an hRSV F-encoding plasmid before labeling with anti-F and anti-GM130, a marker of the cis-Golgi network, 24 h post-transfection. As shown in Figure 2B, the expression of F alone induced the formation of syncytia similar to those observed during infection, and the treatment with Retro-2.2 at 2 µM impaired the formation of large syncytia. F was also observed in perinuclear granules distinct from the cis-Golgi and at the plasma membrane in both conditions. It is noteworthy that no modification of the GM130 labeling was observed in the presence of Retro-2.2.

These data suggest that the inhibition of large syncytia formation upon Retro-2.2 treatment during hRSV infection is mainly dependent on its effect on the F protein.

### 2.3. Retro-2.2 Affects the Trafficking of the F Protein to the Plasma Membrane

The F protein is a class I fusion protein expressed through the endoplasmic reticulum and Golgi network as a precursor F0. It is glycosylated, cleaved by furine to generate the F1 (50 kDa) and F2 (15 kDa) subunits linked by two disulphide bonds, and forms trimers folded in a prefusion conformation [37,38,39] (Figure 3A). We therefore investigated if the Retro-2.2 treatment could impair the maturation of F and/or its trafficking to the plasma membrane. We first analyzed the migration profile on SDS-PAGE of the F protein from infected cells incubated in the absence or presence of Retro-2.2 for 48 h. Cells were lysed in reducing and non-reducing condition to observe the F1 subunit only or F1 linked to F2, respectively. Surprisingly, Western blot analysis revealed similar migration profiles in all the conditions, with only a band of approximatively 50 kDa corresponding to the F1 subunit (Figure 3B). Although we were not able to detect the full-length form of F in these conditions, these data show that Retro-2.2 did not abrogate the cleavage by furine. However, the treatment with Retro-2.2 led to a lower quantity of F protein, which correlates with the inhibition of viral spreading. Nevertheless, we cannot exclude a partial defect of F maturation or the induction of F degradation upon Retro-2.2 treatment. In order to assess the effect of Retro-2.2 on F trafficking, we then quantified the presence of F at the plasma membrane. To overcome the potential bias linked to cell fusion and of immunostaining, we generated an F surrogate which consisted in the insertion in place of the F ectodomain of the sequences of the mCherry and of the HiBit peptide, an 11-amino-acid-long optimized peptide of the split Nano Luciferase (Figure 3A) [40,41,42]. The plasmid encoding this chimeric protein was transfected in HEp-2 cells which were incubated for 24 h in the absence or presence of 2 µM Retro-2.2. Analysis of mCherry by fluorescence revealed that the chimeric protein mainly concentrated within intracellular granules upon Retro-2.2 treatment compared to the control condition (Figure 3C, upper panel). Western blot analysis of the chimeric protein expression revealed with an anti-mCherry antibody allowed to confirm that Retro-2.2 did not affect cell transfection nor protein expression or stability (calculated expected size, 42 kDa) (Figure 3C, lower panel). In order to quantify F expression in cells and its specific localization at the plasma membrane, the luminescence activity in total cell lysates or in the cell culture medium was then measured. Our data revealed similar levels of total luminescence in both conditions but a clear decrease in extracellular luminescence induced by Retro-2.2 treatment (Figure 3D). Finally, we performed a similar experiment to quantify extracellular luminescence in the presence of increased concentrations of Retro-2.2. Our results show a dose-dependent inhibition of the presence of the chimeric protein at the plasma membrane, with a calculated IC_50_ close to 1.3 µM (Figure 3E), which correlated with the antiviral IC_50_ of Retro-2.2 (Figure 1A).

Altogether, these results show that Retro-2.2 impairs the trafficking of F to the plasma membrane.

### 2.4. Impact of Retro-2.2 on the G Glycoprotein Trafficking to the Plasma Membrane

We then studied whether the antiviral activity of Retro-2.2 against hRSV was restricted to F or could also modify the trafficking of the G glycoprotein to the plasma membrane. The G glycoprotein is expressed as a type II membrane protein (Figure 4A), but it can also be secreted as a truncated form [43,44]. Although the transmembrane G protein is involved in the initial attachment to the host cell membrane during entry, the G protein was shown to be dispensable for hRSV replication in cell culture [45]. Therefore, the potential effect of Retro-2.2 on this protein could not have been observed using our previous assays. Using a similar strategy described above for F quantification, we generated a chimeric protein by replacing the extracellular domain of G by the HiBit sequence in fusion with EGFP (Figure 4A) and performed the assay described above. Observation of HEp-2 cells transfected with this construct for 24 h in the absence or presence of Retro-2.2 revealed the presence of chimeric protein within intracellular compartments, but no chimeric protein was clearly detected at the plasma membrane (Figure 4B). However, quantification by luminescence of the chimeric protein expression allowed to validate its presence at the plasma membrane and revealed that Retro-2.2 treatment induced a decrease in the quantity of the chimeric protein at the plasma membrane (Figure 4C). Of note, the expression of the chimeric protein was similar in cells treated with Retro-2.2 compared to the control condition, as confirmed by total luminescence quantification and Western blot analysis of the chimeric protein expression revealed with anti-GFP antibody (calculated expected size, 38 kDa) (Figure 4C). Our results also showed a dose-dependent inhibition of the presence of this chimeric protein at the plasma membrane upon Retro-2.2 treatment, with a calculated IC_50_ close to 1.1 µM (Figure 4D).

These results suggest that Retro-2.2 could affect not only F but also G trafficking to the plasma membrane.

### 2.5. Down-Expression of Syntaxin-5 and Sec16A Impair hRSV Replication and Syncytia Formation

We then considered whether Retro-2.2 antiviral activity could be related to the mechanism of action already described for the Retro-2 compounds. The cellular protein Sec16A, which is involved in the endoplasmic reticulum secretion pathway, was previously identified as a target of the Retro-2 compound [46]. More specifically, Sec16A is a protein located at the reticulum endoplasmic exit sites that regulates the formation of COPII vesicles. Unfortunately, in the absence of efficient antibodies against Sec16A, we did not succeed to study the impact of Retro-2.2 treatment on this protein. The Syntaxin-5 (Syn5), a SNARE protein that is involved in cellular retrograde transport pathways, was also previously described to be relocated in the presence of Retro-2 [28]. We therefore assessed the impact of Retro-2.2 treatment on Syn5 expression. Uninfected and hRSV-infected HEp-2 cells (MOI 0.2) were incubated for 48 h in the presence of Retro-2.2 at 2 µM, before analysis of cell lysates by Western blot. As shown in Figure 5A, Retro-2-2 treatment induced a decrease in Syn5 expression in both uninfected and infected cells. In parallel, non-infected cells incubated in the same condition were fixed, before immunolabeling of Syn5. In the absence of treatment, Syn5 was shown to mainly concentrate in the Golgi apparatus and to present a diffuse cytoplasmic localization expected to correspond to the endoplasmic reticulum (Figure 5B). Upon Retro-2.2 treatment, Syn5 presented only a diffuse cytoplasmic localization (Figure 5B). These results confirmed previous observation concerning the relocalization of Syn5 upon Retro-2 treatment [28] but also highlighted that Retro-2.2 treatment affects Syn5 expression. These data also suggest that the effect of Retro-2.2 on Syn5 could be responsible for its antiviral activity.

We therefore studied the impact of the down-regulation of Syn5 or Sec16A expression on hRSV replication. Human epithelial A549 cells, which are known to be more permissive to siRNA transfection compared to Hep-2 cells, were transfected with siRNA for 24 h, then infected for 48 h with Luc-rRSV with an MOI of 0.2. Quantification of viral replication by luminescence measurement on cell lysates revealed that down-regulation of Syn5 or Sec16A both induced a 30–40% decrease in the infection (Figure 5C). Western blot analysis of cell lysates confirmed the down-expression of the two isoforms of Syn5 at this final time point (Figure 5D) [47]. However, we did not manage to validate Sec16A extinction as we did not succeed to reveal Sec16A expression by Western blot. In order to observe the impact of siRNA on syncytia formation during infection, the same experiment was performed on cells infected with mCherry-rRSV. As shown in Figure 5E, down-regulation of Syn5 and Sec16A expression impaired the formation of large syncytia compared to the control condition.

Therefore, our data reveal that Retro-2-2 alters the expression of Syn5 and that down- regulation of Syn5 or Sec16A expression affects the capacity of hRSV to form large syncytia in cell culture. Of note, hRSV replication is less impacted by the extinction of Syn5 or Sec16A alone by siRNA than by the treatment by Retro-2.2. These observations suggest that Retro-2.2 viral inhibition could be linked to its inhibitory activity on the retrograde pathway.

## 3. Discussion

Although vaccines against hRSV have been recently commercialized, there is still a need for antiviral treatments to treat hRSV infections. Many attempts have been made by pharmaceutical companies to develop compounds specifically targeting the viral F and L proteins [48,49,50]. However, the emergence of escape mutants upon treatment highlights the necessity to search for alternatives and/or complementary antiviral approaches. Among those, strategies aiming at interfering with intracellular trafficking pathways required for pathogen infections could be promising.

In the present study, we evaluated the potential antiviral activity of the Retro-2.2 compound. This compound was derived from the structure–activity relationship study performed to increase the efficiency of the initial Retro-2 molecule and its optimized Retro-2.1 analogue (EP3886859). Retro-2.2 was found to be twice more effective against Shiga toxins than Retro-2.1, Shiga toxins being reference bacterial toxins using the retrograde transport in our structure–activity relationship study. Retro-2 and Retro-2.1 were shown to be active against a large panel of viruses [32]. Here, we first showed that Retro-2.2 presents an antiviral activity against hRSV in cell cultures, in the micromolar range. However, we also detected a non-expected cytotoxic effect of the compound in our conditions, with a CC_50_ close to 15 µM. This low selectivity index value compared to the one obtained for the optimized Retro-2.1 molecule in previous studies [51] will need to be investigated.

Our data also show that Retro-2.2 did not impact the viral particle or the initial infection nor the formation of viral factories, but it induced a defect of viral particles release associated to a decrease in syncytia formation in cell culture. Altogether, these results suggest that Retro-2.2 targets a late stage of the viral cycle, and more specifically the F protein function. Of note, the titration of the virus produced from cells treated with Retro-2.2 revealed that new virions formed only small plaques, suggesting that Retro-2.2 treatment could also induce a defect of virions morphogenesis. By engineering chimeric transmembrane fluorescent proteins which display the HiBit peptide fused with the transmembrane sequences of the F or G proteins, we were able to reveal that Retro-2.2 could impair the traffic of the F but also of the G protein to the plasma membrane. However, despite several attempts of co-immunofluorescence labeling of F and proteins of either the endoplasmic reticulum, the Golgi apparatus or the endosomes, we were not able to characterize a specific site of retention of the F and G proteins inside the cells. As the initial compound Retro-2 was shown to affect the cellular proteins Sec16A and Syn5, we therefore assessed if the antiviral activity of Retro-2.2 against hRSV could be due to its action on these proteins. Although we did not manage to investigate the impact of Retro-2.2 treatment on Sec16A, our results show a decrease in Syn5 expression, with relocalization of this protein upon Retro-2.2 treatment. In parallel, we showed that down-regulation of Syn5 and Sec16A also inhibited hRSV replication in cell culture and the formation of large syncytia. Taken together, our results strongly suggest that, by modifying the cellular retrograde pathway, Retro-2.2 affects the trafficking of hRSV membrane proteins to the plasma membrane (Figure 6). However, we cannot exclude that Retro-2.2 treatment could also alter the full maturation of the F and G proteins and virions morphogenesis. Noteworthy, in order to decipher the specificity of action of Retro-2.2 against hRSV, and to gain information on the cytotoxicity of the treatment, it would be necessary to evaluate the impact of Retro-2.2 treatment on the trafficking of cellular proteins known to be addressed to the plasma membrane such as TGN46 or Cl-MPR.

As mentioned above, Retro-2.2 displays only low antiviral activity in cell culture. Nevertheless, given the dual effect of the treatment on F and G proteins, and based on the fact that G protein is not critical for hRSV replication in cell culture but favors in vivo replication, it would be interesting to test the antiviral activity of this compound in vivo in a mouse model. For this purpose, one would expect that, as recently shown for Retro-2.1, the formulation in a thermosensitive hydrogel could foster the delivery of Retro-2.2 in vivo [52].

In conclusion, our data highlight that molecules that interfere with the trafficking of intracellular vesicles could be used as antivirals against hRSV. Optimization of such molecules to improve their selectivity index, their solubility, and their delivery in vivo still remains necessary. Finally, our results allowed to extend the list of the pathogens sensitive to Retro-2 derivatives and reinforce the potential interest of these molecules as broad-spectrum antivirals.

## 4. Materials and Methods

### 4.1. Cells

HEp-2 cells (ATCC, CCL-23), A549 (ATCC, CCL-185) and 293T cells were maintained in Dulbecco’s modified Eagle’s medium (DMEM; Eurobio, Les Ulis, France) supplemented with 10% heat-inactivated fetal calf serum (FCS; Eurobio, Paris, France), 1% L-glutamine, and 1% penicillin–streptomycin. The transformed human bronchial epithelial cell line (BEAS-2B) (ATCC CRL-9609) was maintained in RPMI 1640 medium (Eurobio) supplemented with 10% FCS, 1% L-glutamine, and 1% penicillin–streptomycin. The cells were grown at 37 °C in 5% CO_2_ and transfected using Lipofectamine 2000 (Invitrogen, Whaltham, MA, USA), as described by the manufacturer.

### 4.2. Retro-2.2 Compound Synthesis

A mixture containing 6-fluoro-1-methyl-2*H*-benzo[*d*][1,3]oxazine-2,4(1*H*)-dione (500 mg, 2.56 mmol), 2-aminophenol (280 mg, 2.56 mmol), and 5-(2-methyl-1,3-thiazol-4-yl)-2-thiophenecarbaldehyde (536 mg, 2.56 mmol) in acetic acid (3 mL) was heated at 130 °C in a microwave synthesizer (CEM) for 4 h. The crude mixture was diluted in ethyl acetate, and a saturated NaHCO_3_ aqueous solution was added. The mixture was stirred vigorously for 1 h. The aqueous layer was extracted two more times with EtOAc. The combined organic phase was dried over magnesium sulfate, filtered off, and the solvent was evaporated under vacuum. Purification by column chromatography (CombiFlash Serlabo; column 40 G) on silica gel using cyclohexane/ethyl acetate (60/40) afforded 1 as a pale brown solid (985 mg, 85% yield). At the end of the synthesis, the compound was analyzed by ^1^H and ^13^C nuclear magnetic resonance (NMR) and mass spectrometry (LC–MS). The *m*/*z* calculated for C_23_H_18_FN_3_O_2_S_2_H [M+H+]^+^ (Retro-2.2) was 452.4 Da, with a purity of 99.5%.

### 4.3. Viruses and Titration by Plaque Formation Assay

The recombinant mCherry-rRSV and Luc-rRSV corresponding to the RSV Long strain expressing either the mCherry or the firefly luciferase protein were amplified in HEp-2 cells and titrated using a plaque assay procedure as previously described [36]. Briefly, for titration, cells were infected with serial 10-fold dilutions of the viral supernatant in complete minimum essential medium (MEM). The overlay was prepared with Avicel RC581 microcrystalline cellulose (FMC Biopolymer, Philadelphia, PA, USA) at a final concentration of 0.6% in complete MEM containing 1% fetal calf serum. After 6 days at 37 °C with 5% CO_2_, plaques were revealed by staining of the cell layers with a solution containing 0.5% crystal violet and 20% ethanol, and the number of PFUs per well was counted.

### 4.4. Plasmids

pcDNA3.1 codon-optimized plasmid for mammalian expression encoding the hRSV F was a gift from Marty Moore, Emory University [53]. The sequences encoding the chimeric proteins (primary sequences shown in Figure 3 and Figure 4) were optimized for eukaryotic expression and commercially made (GeneCust, Boynes, France). The full-length sequence of mCherry-HiBit-F was cloned into pCI vector (Genbank, Bethesda, MD, USA, accession n° U47119). For HiBit-GFP-G expression, the optimized sequence encoding the N-terminal and transmembrane domains of G, in fusion with the HiBit sequence, was cloned in fusion with EGFP into the pEGFP-N2 vector (Clontech, New Montain, CA, USA).

### 4.5. Antibodies

The following primary antibodies were used for immunostaining and/or immunoblotting: mouse anti-F (BioRad, Hercules, CA, USA), mouse anti-F 2F7 (Abcam, Cambridge, UK), mouse anti-G (Abcam), rabbit anti-N antiserum [54], rabbit anti-GM130 (Abcam), mouse anti-Syntaxin-5 (Santa Cruz, Dallas, TX, USA), mouse monoclonal anti-β tubulin antibody (Sigma-Aldrich, Saint-Louis, MO, USA), rabbit anti-GFP (Invitrogen, Whaltham, MA, USA), and anti-mCherry (Rockland immunochemicals, Limerick, ME, USA).

Secondary antibodies directed against mouse and rabbit IgG coupled to HRP (SeraCare, Milford, CT, USA) were used for immunoblotting, and antibodies directed against mouse and rabbit IgG coupled to Alexa 594 or Alexa 488 (Invitrogen), respectively, were used for immunostaining.

### 4.6. Antiviral and Cytotoxic Activity Assays

Cells were seeded at 5 × 10^4^ cells per well in 96-well plates. Twenty-four hours later, the cells were infected at a multiplicity of infection (MOI) of 0.2 for 2 h with mCherry-rRSV or Luc-rRSV diluted in MEM without phenol red and without FCS. In parallel, Retro-2.2 compound was serially diluted in dimethyl sulfoxide (DMSO) and then further diluted in MEM without phenol red medium containing 2% FCS. Two hours post-infection, the cell culture medium was replaced by the medium containing Retro-2.2, and plates were incubated at 37 °C for 24 to 48 h before analysis. The mCherry fluorescence was measured using a spectrophotometer (Tecan Infinite M200PRO, Männedorf, Switzerland) with excitation and emission wavelengths of 580 and 620 nm, respectively. For Luc-rRSV replication quantification, cells were lysed in luciferase lysis buffer (30 mM of Tris [pH 7.9], 10 mM of MgCl2, 1 mM of DTT, 1% Triton X-100, and 15% glycerol). After the addition of luciferase assay reagent (Promega, Charbonnières, France), luminescence was measured using a Tecan Infinite M200 Pro luminometer (Tecan, Männedorf, Switzerland). The values obtained for non-infected HEp-2 cells were used as a standard for the fluorescence and luminescence background levels, respectively, and the value obtained for infected and untreated cells was used to normalize the data. Cytotoxicity assays were performed on non-infected cells incubated for 48 h with the serial dilutions of Retro-2.2, using the CellTiter-Glo luminescent cell viability kit (Promega). Each experiment was performed in duplicate and repeated at least twice. IC_50_ and CC_50_ values were calculated by fitting the data to a sigmoidal curve equation in GraphPad Prism 6 software (Boston, MA, USA).

### 4.7. siRNA Experiments

The control siRNA and siRNAs against Syn-5 and Sec16A were purchased from Qiagen (Quiagen, Hilden, Germany). Each siRNA is a pool containing 4 different sequences (Catalog numbers: SI03075345, SI03052966, SI00048636, and SI00048615). A549 cells were transfected with the indicated siRNAs at a final concentration of 20 nM by reverse transfection into 96-well plates (7 × 10^4^ cells per well) using Lipofectamine RNAiMAX (Thermo Fisher, Waltham, MA, USA), according to the manufacturer’s instructions. Briefly, a mixture containing Opti-MEM (Invitrogen), Lipofectamine RNAiMAX, and siRNA was incubated for 5 min at room temperature before being deposited at the bottom of the wells. The cells in MEM without antibiotics were then added dropwise before incubation at 37 °C with 5% CO_2_. After 24 h of transfection in the presence of siRNA, the medium was removed, and the cells were infected following the protocol described above. Forty-eight hours post-infection, the mCherry fluorescence was measured using a spectrophotometer (Tecan Infinite M200PRO) with excitation and emission wavelengths of 580 and 620 nm, respectively. The values obtained for non-infected HEp-2 cells were used as a standard for the fluorescence background level, and the values obtained for infected cells transfected with the siCT were used to normalize the data. Each experiment was performed in quadruplicate and repeated three times. For each experiment, cells were finally lysed in Laemmli, and Syn5 and tubulin expression was analyzed by Western blotting.

### 4.8. SDS-PAGE and Western Blot Analysis

Protein samples were separated by electrophoresis on 12% polyacrylamide gels in Tris-glycine buffer. All samples were boiled for 3 min prior to electrophoresis. Proteins were then transferred to a nitrocellulose membrane (BioRad). The blots were blocked with 5% nonfat milk in PBS Tween20 0.2%, followed by incubation with mouse anti-F antibody (1:1000), rabbit anti-N antiserum (1:5000), mouse anti-tubulin (1:1000), or mouse anti-Syn5 (1:2000), and horseradish peroxidase (HRP)-conjugated anti-mouse or anti-rabbit (1:10,000) secondary antibodies (SeraCare). Western blots were developed using Clarity^TM^ Western ECL substrate (Bio-Rad) and exposed using the Bio-Rad ChemiDoc Touch Imaging System (Bio-Rad, Hercules, CA, USA).

### 4.9. Immunofluorescence

Cells were seeded in 24-well plates containing coverslips. Plates were infected or transfected depending on the assay. Cells were fixed using 4% paraformaldehyde for 30 min at room temperature, then permeabilized in PBS buffer containing 3% bovine albumin serum (BSA) and 0.1% Triton X-100 for 30 min at room temperature. Cells were then washed once in PBS-3% BSA-0.05% Tween20 and incubated with relevant mouse or rabbit primary antibody (anti-F 1:1000, anti-N 1:20,000; anti-GM130 1:1000) diluted in PBS-3% BSA-0.05% Tween20 for 1 h at room temperature. Cells were washed 3 times with PBS-3% BSA-0.05% Tween20, then incubated with goat anti-mouse or goat anti-rabbit secondary antibody coupled to Alexa 488 or Alexa 594, respectively. Cells were then washed 3 times with PBS-3% BSA-0.05% Tween20 and two times with PBS. For nucleus labeling, cells were exposed to Hoechst 33,342 stain (Invitrogen) during incubation with secondary antibodies. Coverslips were mounted with ProLong Gold antifade reagent (Invitrogen). Images were obtained using an inverted SP8x confocal microscope (Leica, Welzlar, Germany) using a 63× oil immersion objective. Maximum projections of optical Z slices are shown. Images were processed with Leica imaging software LAS X 1.4.5 and ImageJFIJI software version 2.3.0.

### 4.10. HiBit Luminescence Assay

HEp-2 cells were seeded at 5 × 10^4^ cells per well in 96-well plates or at 2 × 10^5^ cells per well in 24-well plates containing coverslips. Cells were transfected in Opti-MEM (Gibco, Thermo-Fisher) with 0.2 µg of DNA per well (96-well plate) or 1 µg of DNA per well (24-well plate) using Lipofectamine 2000 for 5 h. The medium was then changed for Opti-MEM supplemented with Retro-2.2 at various concentrations. Twenty-four hours post-transfection, the expression of chimeric proteins in total cell lysates or at the cell surface was quantified by luminescence measurement using the Nano-Glo HiBiT Lytic Detection System (Promega, N3030) or the Nano-Glo HiBiT Extracellular Detection System (Promega, N2420), respectively, according to manufacturer’s protocol. For each experiment, cells were finally lysed in Laemmli, and the expression of chimeric proteins was analyzed by Western blotting.

### 4.11. Statistical Analysis

A nonparametric Mann–Whitney test (comparison of two groups) (*n* = 4 or 6) was used to compare unpaired values (GraphPad Prism 6 software). Significance is represented in the figures (*, *p* < 0.05; **, *p* < 0.01).

## Figures and Tables

**Figure 1 ijms-25-00415-f001:**
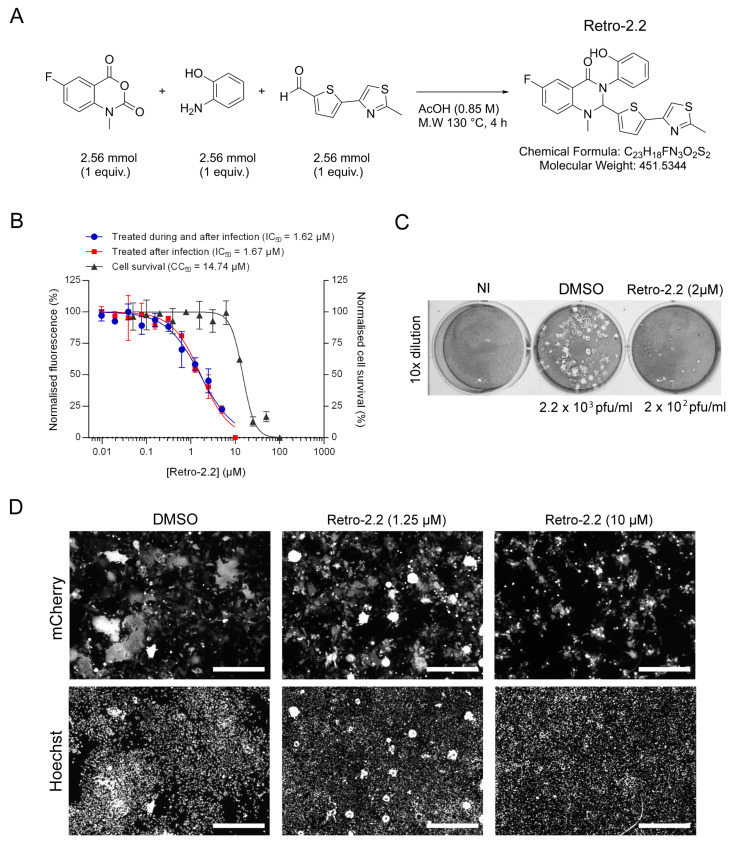
Impact of Retro-2.2 treatment on hRSV replication on Hep-2 cells. (**A**) Scheme of the synthesis of Retro-2.2 compound. (**B**) Cells were infected for 2 h with mCherry-rRSV at MOI 0.2, in the presence (blue curve) or the absence (red curve) of serial dilutions of Retro-2.2. The medium was then replaced to incubate cells in the presence of serial dilutions of Retro-2.2 for 48 h. The viral replication was quantified by measurement of the mCherry fluorescence. In parallel, cell viability upon treatment with Retro-2.2 was quantified in non-infected HEp-2 cells (black curve). Error bars are standard deviations from duplicates. Data are representative of three experiments. The curves were fitted in GraphPad Prism 6 software using a four-parameter logistic (4PL) regression. IC_50_ and CC_50_ are indicated. (**C**) Titration of the virions released into the culture media (diluted 10 times) from non-infected cells (NI) or cells infected with mCherry-rRSV, and treated with DMSO (control) or Retro-2.2 at 2 µM, by plaque formation assay. Calculated viral titers are indicated in pfu/mL. (**D**) Representative images of HEp-2 cell cultures infected with mCherry-rRSV at MOI 0.2, 48 h post-infection in the absence or presence of Retro-2.2 at 1.25 and 10 µM. Upper panel, mCherry fluorescence of infected cells; lower panel, nuclei colored with Hoechst. Scale bars, 500 µm.

**Figure 2 ijms-25-00415-f002:**
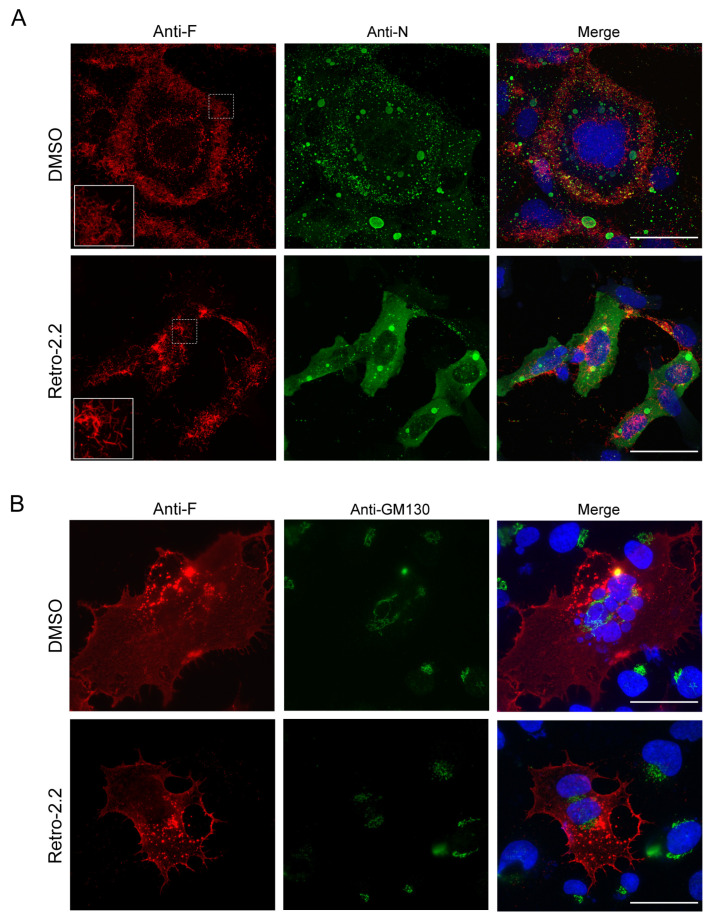
Effect of Retro-2.2 treatment on the localization of N and F viral proteins. (**A**) BEAS-2B cells were infected for 2 h with Luc-rRSV, and the medium was changed to incubate cells in the absence (DMSO) or the presence of Retro-2.2 at 2 µM. Thirty-six hours post-infection cells were fixed, immunostained with anti-F (red) and anti-N (green) antibodies, and nuclei were colored with Hoechst. Insets correspond to the zoom of the regions indicated by dashed squares. (**B**) BEAS-2B cells were transfected with a plasmid encoding the F protein. Twenty-four hours post-transfection cells were fixed and immunostained with anti-F (red) and anti-GM130 (green) antibodies, and nuclei were colored with Hoechst. Images are from confocal microscopy sections. Scale bars, 50 µm.

**Figure 3 ijms-25-00415-f003:**
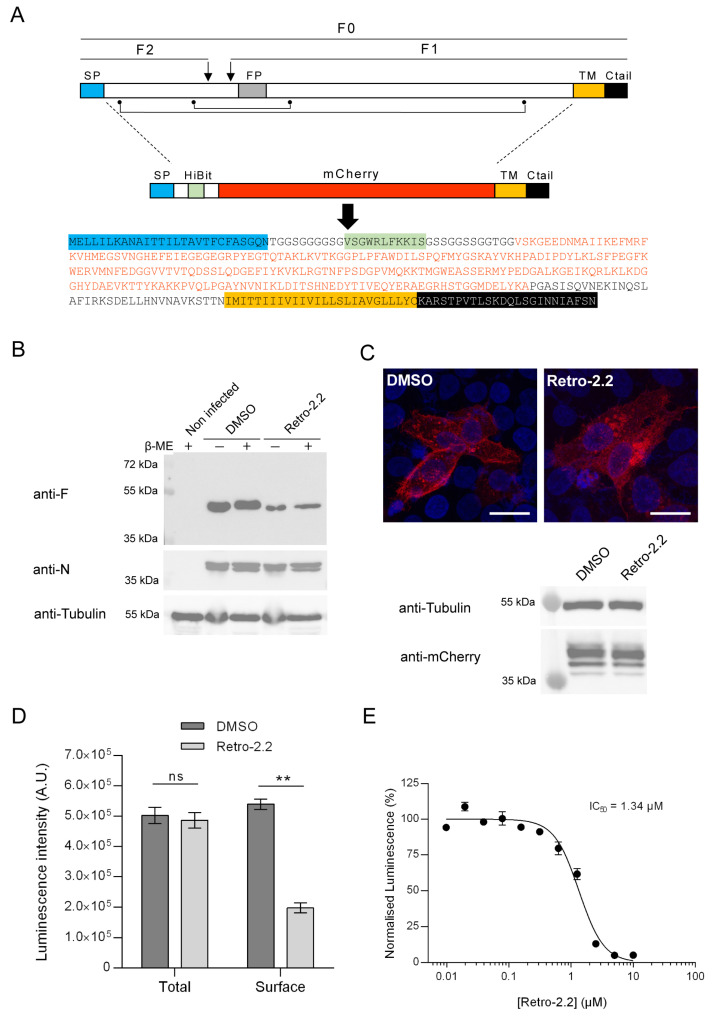
Impact of Retro-2.2 on the trafficking of the F protein to the plasma membrane. (**A**) Schematic representation of the F primary sequence and of the derived chimeric HiBit-mCherry-F protein. The F0 form and the F1 and F2 subunits of F are indicated. Arrows indicate the cleavage sites on F0. The points and lines indicate the cysteines and disulfide bonds. SP, signal peptide; FP, fusion peptide; TM, transmembrane domain; Ctail, intracellular C-terminal domain. The amino acid sequence of the chimeric protein is indicated with domains as colored boxes, and the mCherry sequence is written in red. (**B**) Western blot analysis of the migration profile of the F protein from infected HEp-2 cell lysates incubated in the presence of DMSO or Retro-2.2 (2 µM) during 48 h. The presence or absence of β-Mercaptoethanol (β-ME) is indicated. Membranes were sequentially incubated with anti-F, anti-N, and anti-tubulin antibodies. Mass weights are indicated on the left. (**C**) Hep-2 cells were transfected with the plasmid encoding the chimeric HiBit-mCherry-F protein. Six hours post-transfection, the medium was replaced to incubate the cells in the presence of Retro-2.2 at 2 µM. Twenty-four hours post-transfection, cells were fixed, and nuclei were stained with Hoechst (blue), before analysis of the mCherry fluorescence (red) by confocal microscopy (**upper panel**). Scale bars, 20 µm. The expression of the chimeric protein was also validated by Western blot using an anti-mCherry antibody (**lower panel**) and anti-tubulin as a control. (**D**) In parallel, the presence of the chimeric protein in whole-cell lysates or at the cell surface was quantified by luminescence measurement. Data are means SEM from quadruplicates. **, *p* < 0.01, ns, not significant. Data are representative of three experiments. (**E**) A similar experiment was performed in the presence of serial dilutions of Retro-2.2, and the presence of the chimeric protein at the cell surface was quantified by extracellular luminescence measurement. Error bars are standard deviations from duplicates. Data are representative of two experiments. The curves were fitted in GraphPad Prism 6 software using a four-parameter logistic (4PL) regression. IC_50_ is indicated.

**Figure 4 ijms-25-00415-f004:**
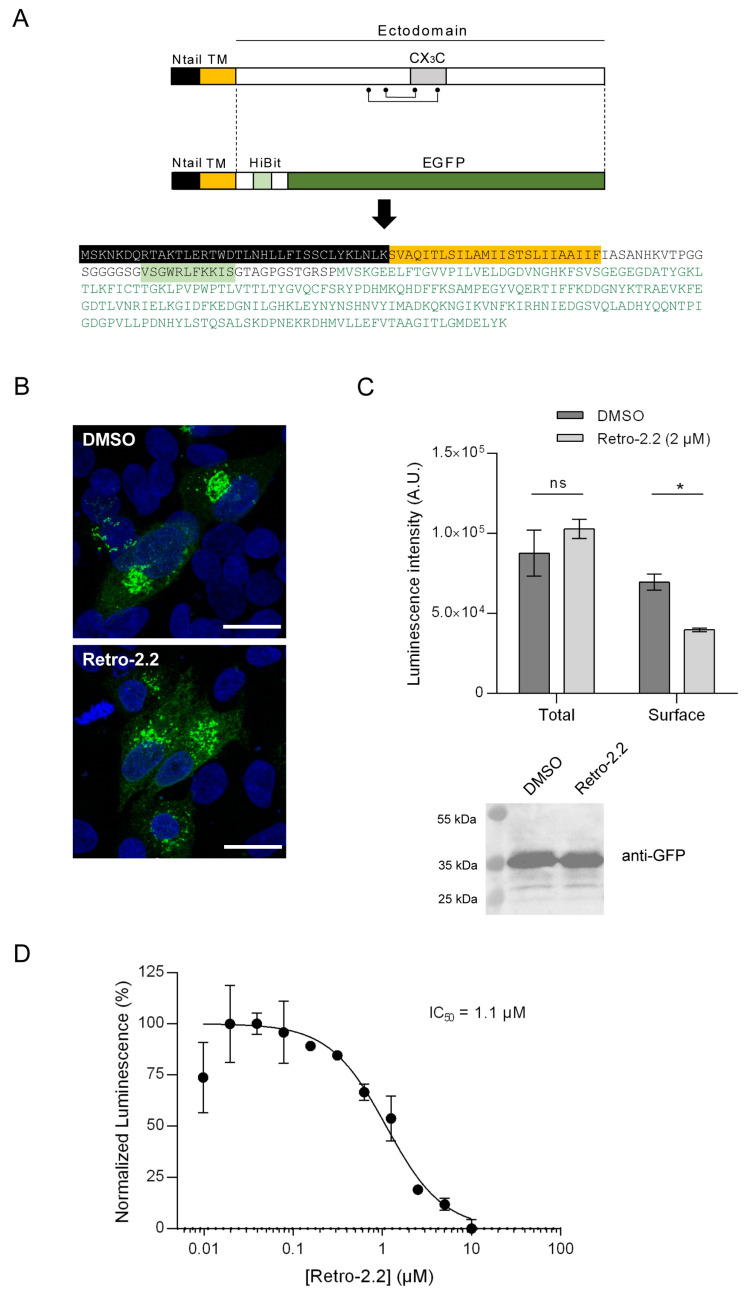
Impact of Retro-2.2 on the addressing of the hRSV G protein to the plasma membrane. (**A**) Schematic representation of the G protein and of the derived chimeric HiBit-EGFP-G protein. The points and line indicate the cysteines and disulfide bonds. Ntail, intracellular N-terminal domain; TM, transmembrane domain; CX_3_C, binding domain on the CX_3_C receptor. The amino acid sequence of the chimeric protein is indicated with domains represented as colored boxes, and the EGFP sequence is written in green. (**B**) HEp-2 cells were transfected with the plasmid encoding the chimeric HiBit- EGFP-G protein. Six hours post-transfection, the medium was replaced to incubate the cells in the presence of Retro-2.2 at 2 µM. Twenty-four hours post-transfection, cells were fixed and nuclei were stained with Hoechst (blue), before analysis of the EGFP fluorescence (green) by confocal microscopy. Scale bars, 20 µm. (**C**) In parallel, the luminescence in whole-cell lysates or in the extracellular environment was quantified (**upper panel**), and expression of the chimeric protein was validated by Western blot using an anti-GFP antibody (**lower panel**). Data are means SEM from quadruplicats. *, *p* < 0.05; ns, not significant. Data are representative of two experiments. (**D**) A similar experiment was performed in the presence of serial dilutions of Retro-2.2, and the presence of the chimeric protein at the cell surface was quantified by extracellular luminescence measurement. Error bars are standard deviations from duplicates. Data are representative of two experiments. The curves were fitted in GraphPad Prism 6 software using a four-parameter logistic (4PL) regression. IC_50_ is indicated.

**Figure 5 ijms-25-00415-f005:**
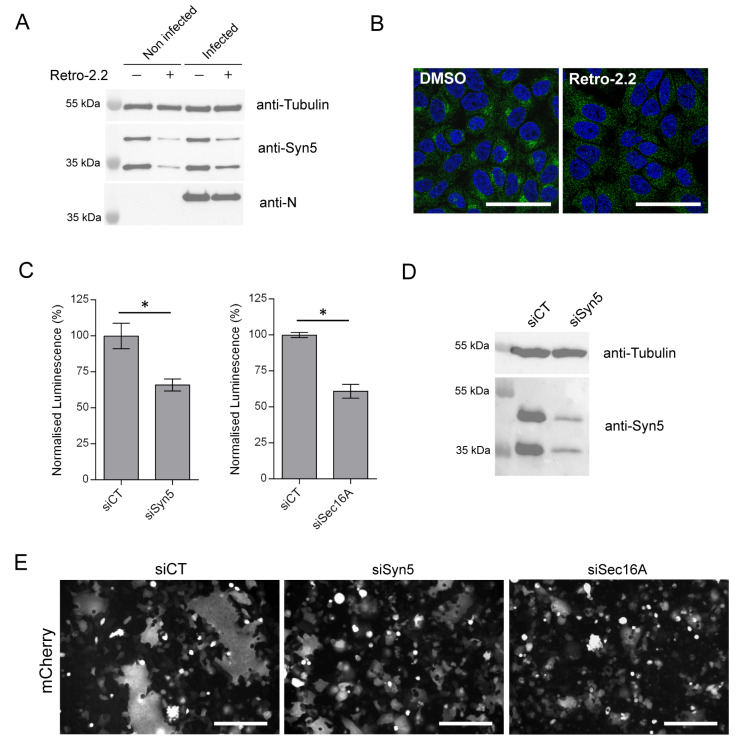
Down-regulation of Syn5 and Sec16A expression impairs hRSV replication and syncytia formation. (**A**,**B**) HEp-2 cells, either non-infected or infected with mCherry-rRSV at MOI 0.2, were incubated for 48 h in the absence (DMSO) or the presence of Retro-2.2 at 2 µM. (**A**) Expression of Syn5 in cell lysates was analyzed by Western blot. (**B**) Non-infected cells were fixed 48 h post-incubation in the presence of Retro-2.2 and immunostained with anti-Syn5 antibody (green), and nuclei were colored with Hoechst (blue). Images are from confocal microscopy sections. Scale bars, 50 µm. (**C**–**E**) A549 cells were transfected with control siRNA (siCT) or siRNA targeting syntaxin-5 (siSyn5) or Sec16A (siSec16A) during 24 h, before infection with Luc-rRSV at MOI 0.2 during 48 h. (**C**) Viral replication was quantified by measuring the luminescence in cell lysates. Data are means SEM from quadruplicates. *, *p* < 0.05. Data are representative of three experiments. (**D**) Western blot analysis of Syn5 expression in total cell lysates. (**E**) A549 cells transfected with siRNA as described above were infected with mCherry-rRSV at MOI 0.2 during 48 h. Representative images of mCherry fluorescence of infected A549 cell cultures. Scale bars, 200 µm.

**Figure 6 ijms-25-00415-f006:**
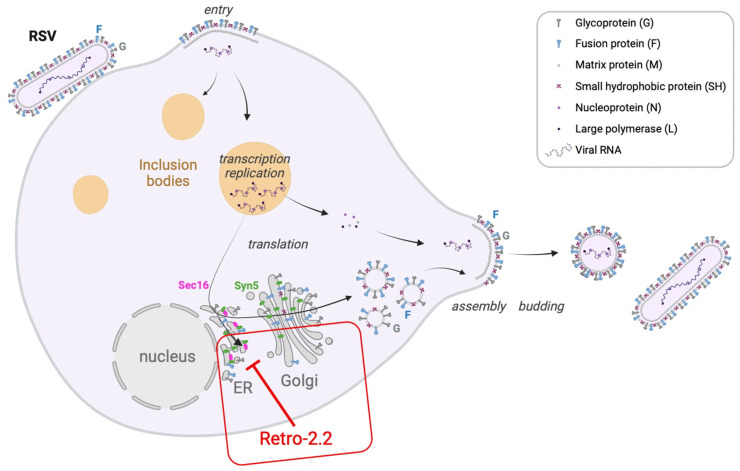
Schematic representation of hRSV replication cycle and of the activity of Retro-2.2. Created with BioRender.com.

## Data Availability

Data is contained within the article and Appendix A.

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
