# Peer review of "A New Derivative of Retro-2 Displays Antiviral Activity against Respiratory Syncytial Virus"

_ijms, 2023, doi:10.3390/ijms25010415_

Round 1
Reviewer 1 Report
Comments and Suggestions for Authors
The article “A new derivative of Retro-2 displays antiviral activity against Respiratory Syncytial Virus” describes the antiviral activity of retro 2.2, a new derivative of the cellular retrograde transport inhibitor Retro-2, against hRSV. The authors report for the first time that this type of inhibitor can reduce hRSV proliferation in a cell-based assay and show an associated reduction of the size of syncytia upon hRSV infection. The authors demonstrate that transport of viral proteins F and G to the cellular membrane is inhibited in a dose dependant manner, and they suggest this transport inhibition is responsible for the antiviral effects. Indeed, retro 2.2 does not inhibit virus binding or fusion (early stage of virus lifecycle), as well as viral proteins expression level and proteolytic activation. It only impacts viral envelop proteins transport to the membrane of infected cell (inhibition of virus lifecycle late stage). However, the authors are reporting that retro 2.2 has a low selectivity index and emphasise that more molecular optimisation is needed to produce and deliver an inhibitor of the intracellular vesicles trafficking to the infected cells.
Overall, the article is very well written and structured in a logical way. Methods used have been appropriate to test the proposed hypothesis and authors are fully aware of the limitations of retro 2.2 as antiviral molecule as it is at the moment. However, I would recommend the authors to provide additional information on the effect of retro 2.2 on cell proliferation (non-infected) and transport of endogenous membrane proteins to the cell membrane (non-treated vs. treated, and non-infected vs. infected).
Indeed, cytotoxicity has been evaluated when cell monolayer is already fully confluent and usually confluent cells tend to not proliferate as fast than cells seeded below full confluency. I would suggest the authors to include additional data that describe the effect of retro 2.2 using lower densities for seeding cell (i.e. 10,000 and 5,000 cells per well in a 96-well plate format). This experiment would provide additional information on the impact of retro 2.2 on the overall cell fitness. Indeed, cells must be in full physiological and metabolic capacities to be able to produce normal level of virions progeny.
I would also suggest the authors to evaluate and quantify the transport efficacy of cell membrane proteins using the same transport mechanism than hRSV F and G. Indeed, F and G proteins are most likely over-expressed during infection in comparison to other cellular proteins as the virus is turning the cell machinery to its own advantage. It would be interesting to see if transport of endogenous membrane proteins (non-infected vs. infected cells) is inhibited at a similar level when cells are in presence of retro 2.2 at a concentration close to the IC50 values reported in the article (1.5-3 µM).
Data about cell proliferation and amount of endogenous membrane proteins transported to the membrane in presence of increasing concentration of retro 2.2 would be a nice addition to the article and would provide valuable supplementary information to the reader.
Comments on the Quality of English LanguageOverall, the quality of English is excellent. Only minor correction is suggested:
Use the term “newborn” instead of “new-born” in the document.
Author Response
We thank the reviewer for these positive comments. Based on the reviewer’s suggestions, we performed new experiments to investigate further the toxicity of Retro-2.2.
- Indeed, cytotoxicity has been evaluated when cell monolayer is already fully confluent and usually confluent cells tend to not proliferate as fast than cells seeded below full confluency. I would suggest the authors to include additional data that describe the effect of retro 2.2 using lower densities for seeding cell (i.e. 10,000 and 5,000 cells per well in a 96-well plate format). This experiment would provide additional information on the impact of retro 2.2 on the overall cell fitness. Indeed, cells must be in full physiological and metabolic capacities to be able to produce normal level of virions progeny.
As suggested by the reviewer, the toxicity of Retro-2.2 treatment on cultures of cells plated at different confluence was assessed. The results, presented in the new supplemental Figure 1B revealed that confluent cells are more sensitive to Retro-2.2 treatment compared to non-confluent cells. However, this difference in sensitivity is only visible for Retro-2.2 concentrations above 10 µM.
- I would also suggest the authors to evaluate and quantify the transport efficacy of cell membrane proteins using the same transport mechanism than hRSV F and G. Indeed, F and G proteins are most likely over-expressed during infection in comparison to other cellular proteins as the virus is turning the cell machinery to its own advantage. It would be interesting to see if transport of endogenous membrane proteins (non-infected vs. infected cells) is inhibited at a similar level when cells are in presence of retro 2.2 at a concentration close to the IC50 values reported in the article (1.5-3 µM).
Studying the transport of endogenous membrane proteins is an interesting remark, and one that would require further study. We discussed this point in the discussion section of the first version of the manuscript, l.394-397 of the revised version. However, we believe that performing steady-state and functional experiments on different cargo proteins is a study in itself and cannot be carried out within the scope of our article.
Reviewer 2 Report
Comments and Suggestions for Authors
The manuscript by Le Rouzic et al entitled “A new derivative of Retro-2 displays antiviral activity against Respiratory Syncytial Virus” is well written and organized. The authors performed a series of in vitro experiments to determine whether Retro-2 had anti-viral activity against hRSV and the mechanism involved.
I have some questions I would like the authors to address:
1. Why do the authors use different cell lines to perform the experiments? Figure 1,3 and 4 Hep-2, Figure 2 BEAS-2B, Figure 5 A549.
2. Figure 5 legend says HEp-2 cells while in the text it is written A549 cells. Please, correct.
3. There is ample literature suggesting that Retro-2 can inhibit entry of viruses into cells when treated before infection. The authors should explore this possibility. It will be interesting to know whether Retro-2.2 has a double effect in hRSV, preventing infection and in the later stages of the virus cycle.
3. The connection between syntaxin-5, Sec16A and Retro-2 mechanism of action is non-existent. Sure, knockdown of syntaxin 5 impaired hRSV replication and syncytia formation, but that does not mean it is the mechanism of action of Retro-2. How is the expression and distribution of syntaxin-5 during Retro-2 treatment? Western blot to check the expression and immunocytochemistry to check intracellular distribution will be useful to determine whether syntaxin-5 is modulated by Retro-2.
4. A schematic showing the virus replication cycle and where Retro-2 is acting will be very helpful
Author Response
We thank the reviewer for these comments and for the experiments suggested to perform in order to improve our manuscript. Based on the reviewer’s suggestions, we performed new experiments, and give a point by point answer to reviewer’s comments.
I have some questions I would like the authors to address:
- Why do the authors use different cell lines to perform the experiments? Figure 1,3 and 4 Hep-2, Figure 2 BEAS-2B, Figure 5 A549.
As mentioned in the manuscript, these three cell lines are all used for hRSV studies, and are sensitive to Retro-2.2 treatment (see Supp Fig.2). However, we agree that we should have explain the specific use of these cells depending the experiment performed. If HEp-2 cells were the standard cells used for the study, we used BEAS-2B cells which display bigger cytoplasm and allow a clearer observation of viral proteins localization by immunofluorescence. For siRNA, A549 cells were used due to their higher efficiency of transfection thus of extinction of gene expression.
We justified the choice of these cells in the new version of the manuscript (l.146-147; l.318-319)
- Figure 5 legend says HEp-2 cells while in the text it is written A549 cells. Please, correct.
We thank the reviewer to point this mistake. The legend of the revised Figure 5 has been corrected.
- There is ample literature suggesting that Retro-2 can inhibit entry of viruses into cells when treated before infection. The authors should explore this possibility. It will be interesting to know whether Retro-2.2 has a double effect in hRSV, preventing infection and in the later stages of the virus cycle.
As suggested by the reviewer, we performed new experiments to study the impact of a 3 h treatment of cells in the presence of Retro-2.2 before infection. These results, presented in the new supplemental Figure 1A showed that Retro-2.2 treatment did not improve the efficiency of the treatment. These additional data confirm that Retro-2.2 does not affect virus entry. Of note, we also modified the legend of the graph of Figure 1B in order to clarify the treatment, compared to the graph presented in supplemental Figure 1A.
- The connection between syntaxin-5, Sec16A and Retro-2 mechanism of action is non-existent. Sure, knockdown of syntaxin 5 impaired hRSV replication and syncytia formation, but that does not mean it is the mechanism of action of Retro-2. How is the expression and distribution of syntaxin-5 during Retro-2 treatment? Western blot to check the expression and immunocytochemistry to check intracellular distribution will be useful to determine whether syntaxin-5 is modulated by Retro-2.
We deeply thank the reviewer for this comment. As suggested, we assessed Syn5 expression by Western blot in both uninfected and infected cells treated by Retro-2.2. Our data, presented in the new version of the Figure 5A, show a decrease of Syn5 expression upon Retro-2.2 treatment. In parallel, we studied the localization of Syn5 in untreated and treated cells. As observed in previous studies for Retro-2, the treatment by Retro-2.2 induced a clear relocalization of Syn5 (see new Figure 5B). Altogether, these results suggest that Retro-2.2 affects Syn5 expression and localization. As a consequence, the corresponding paragraph as well as the discussion were revised.
- A schematic showing the virus replication cycle and where Retro-2 is acting will be very helpful
Based on the reviewer’s comment, we added a schematic representation of hRSV replication cycle showing the proposed action of Retro-2.2 in the new version of the manuscript, in figure 6.
Round 2
Reviewer 1 Report
Comments and Suggestions for Authors
The revised manuscript of "A new derivative of Retro-2 displays antiviral activity against Respiratory Syncytial Virus" comprehensively addresses the comments and suggestions raised by both reviewers. The revisions demonstrate a fair consideration of their feedback, and the updated version strengthens the overall contribution of the work.
While reviewer’s point regarding further investigating the role of Retro 2.2 on different cargo proteins transport to the membrane is acknowledged, such a comprehensive analysis necessitates indeed a substantial amount of work which could lead to its own dedicated study.
Based on the substantial improvements made and the additional data presented, I strongly recommend accepting the revised article for publication in its current form (typographical errors being duly considered).
Reviewer 2 Report
Comments and Suggestions for Authors
The authors have answer all my questions.
The manuscript has improved tremendously and it is ready for publication.